

# Effects of research disturbance on nest survival in a mixed colony of waterbirds

Jocelyn Champagnon, Hugo Carré and Lisa Gili

Tour du Valat, Research Institute for Conservation of Mediterranean Wetlands, Arles, France

## ABSTRACT

**Background**. Long-term research is crucial for the conservation and development of knowledge in ecology; however, it is essential to quantify and minimize any negative effects associated with research to gather reliable and representative long-term monitoring data. In colonial bird species, chicks are often marked with coded bands in order to assess demographic parameters of the population. Banding chicks in multi-species colonies is challenging because it involves disturbances to species that are at different stages of progress in their reproduction.

**Methods**. We took advantage of a long term banding program launched on Glossy Ibis (*Plegadis falcinellus*) breeding in a major mixed colony of herons in Camargue, southern France, to assess the effect of banding operation disturbance on the reproductive success of the three most numerous waterbirds species in the colony. Over two breeding seasons (2015 and 2016), 336 nests of Glossy Ibis, Little Egrets (*Egretta garzetta*) and Cattle Egrets (*Bubulcus ibis*) were monitored from a floating blind in two zones of the colony: one zone disturbed twice a year by the banding activities and another not disturbed (control zone). We applied a logistic-exposure analysis method to estimate the daily survival rate (DSR) of nests and chicks aged up to three weeks.

**Results**. Daily survival rate of Glossy Ibis was reduced in the disturbed zone while DSR increased for Little and Cattle Egrets in the disturbed zone. Nevertheless, DSR was not reduced on the week following the banding, thus discarding a direct effect of handling on breeding success of Glossy Ibis. The protocol and statistical analysis presented here are robust and can be applied to any bird species to test for the effect of a research disturbance or other short and repeated temporal events that may affect reproductive success over one or more breeding seasons.

## INTRODUCTION

Human activities including hunting (*Madsen & Fox, 1995*), recreation (*Navedo & Herrera, 2012*), ecotourism (*Klein, Humphrey & Percival, 1995*; *Guillemain et al., 2008*; *Monti et al., 2018*) and even birdwatching (*Kronenberg, 2014*) can have negative impacts on birds. Yet, the potential effects of ornithological research are less well understood. The aim of conservation research studies is generally to contribute to our knowledge about the studied species. When undertaking this research, it is necessary to take into account disturbance by researchers in order to minimize the effects on the studied population (*Carney &*

Corresponding author
Jocelyn Champagnon,
champagnon@tourduvalat.org

*Sydeman, 1999*). Nevertheless, disturbance is rarely assessed numerically and it is expressed in language that is open to interpretation (*Wilson & McMahon, 2006*)

Research activities that may lead to reduced nest survival (*Götmark, 1992*; *Fair, Paul & Jones, 2010*), are particularly a concern as this can influence the outcome of breeding directly through nest desertion (*Tremblay & Ellison, 1979*; *Piatt, Roberts & Hatch, 1990*; *Shealer & Haverland, 2000*; *Criscuolo, 2001*) or indirectly through an elevated risk of suffering nest predation (*Ibáñez Álamo, Sanllorente & Soler, 2012*). Evidences of negative impacts exist but they are largely dependent on taxa and the timing involved. For instance, nest desertion occurs more frequently at an early stage of incubation in Black-crowned night herons (*Tremblay & Ellison, 1979*). However, in a relatively recent study, *Ibáñez Álamo, Sanllorente & Soler (2012)* provide meta-analytical evidence that research-induced disturbances may not necessarily involve increased nest predation.

Marking birds with an uniquely numbered band to study the population dynamics is a common practice (*Spotswood et al., 2012*). Bird banding and other capture activities may induce stress to the handled bird and the surrounding individuals. This has influenced many countries to require the activities to be performed by trained and licensed ornithologists (*Romero & Romero, 2002*; *Balmer et al., 2008*). Despite the risks involved, it is often assumed that studies arising from so called capture-recapture data lead to a wide variety of information surpassing the potential risk to the animals.

Seabirds and coastal waterbirds often nest in colonies (*Kushlan et al., 2002*). Banding chicks in multi-species colonies is challenging because not all species are synchronized, causing disturbances of species with different ecology and sensitivity. There have been recommendations for minimizing investigator's disturbance in this context (*Carney & Sydeman, 1999*). For instance, *Götmark (1992)* recommended to act rapidly to avoid extreme climatic conditions. Nevertheless, to our knowledge, the effect of banding operations in a mixed colony of waterbirds has not been assessed numerically, neither on the breeding success of the species banded, nor on the other species in the colony that are not targeted by the banding activities.

Here we described an experimental study investigating the effects of research-induced disturbance caused by regular banding operations of Glossy Ibis (*Plegadis falcinellus*) chicks located in large, heron-mixed colonies made up of hundreds of breeding pairs. By conducting regular monitoring of the colony from a floating blind that provided non-disturbing entries to the colony (Fig. 1), we were able to assess the effect of research activities on reproductive success.

## MATERIALS & METHODS

### Study area

This study was carried out in a 3 ha heronry situated in an inundated wood of French Tamarisk trees (*Tamarix gallica*), in the Scamandre Natural Regional Reserve, located in western Camargue, in southern France (43°36′27″N, 4°20′44″E) and approved by le Syndicat Mixte pour la protection et la gestion de la Camargue Gardoise. This heronry is currently the largest mixed heron colony in France and one of the largest in Europe

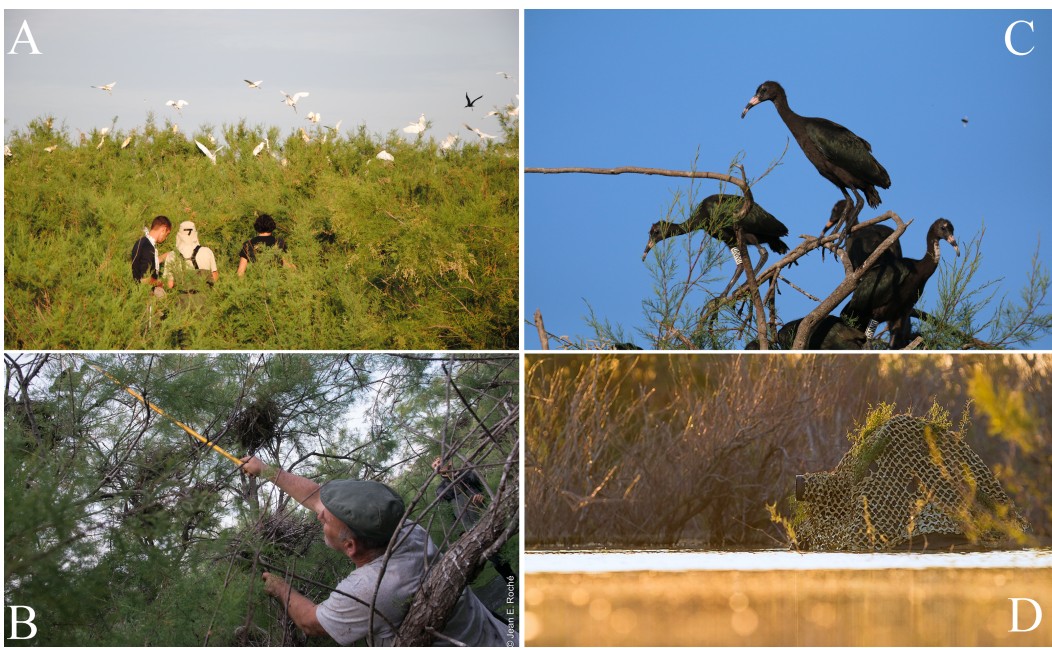

**Figure 1** **Capturing and marking Glossy Ibis chicks in a mixed heron colony.** (A) Flushing adults when entering the colony. Cattle Egrets, Little Egrets, Squacco Herons and Glossy Ibis are identifiable. Photo credit: Tour du Valat. (B) Example of a capture of a Glossy Ibis chick in his nest. Photo credit: Jean-Emmanuel Roché. (C) Glossy Ibis chicks of approximately three weeks old, at the top of the trees. Note that two individuals fit a PVC and metal bands. Photo credit: Jean-Pierre Trouillas. (D) The floating blind used to monitor breeding success. Photo credit: Clément Pappalardo.

(*Gauthier-Clerc, Kayser & Petit, 2006*). Since 2005, the total number of pairs breeding in the heronry (including Glossy Ibis and herons) fluctuated between 3,000 and 9,000 pairs. During 2015, the heronry was composed of 3,700 Cattle Egrets (*Bubulcus ibis*), 3,255 Little Egrets (*Egretta garzetta*), 1,028 Glossy Ibis breeding pairs, 669 Black-crowned night herons, 371 Squacco herons (*Ardeola ralloides*) and 46 Grey herons (*Ardea cinerea*).

## Banding operations

The study took place over two consecutive breeding seasons, between April and July of 2015 and 2016. Two comparable zones of approximately 400 m² were chosen based on their similarity in terms of vegetation density, wind exposure and observer accessibility. One zone was disturbed due to the regular banding operations (disturbed zone), while the other was not (control zone). The two zones were separated by at least 50m of vegetation. Banding operations occurred twice in each season and each operation involved capturing and banding Glossy Ibis chicks between 10 days and three weeks of age (Fig. 1). The operations were conducted early in the morning when meteorological conditions were deemed favorable (i.e., with no wind and a temperature above 15 °C). During each operation, 28 to 36 participants entered the colony and were divided into five to seven teams, each supervised by an official bander (Fig. 1). All participants were briefed on the methods before entering the colony and operations were limited to a maximum duration

of 1 h. During this time, between 123 and 365 chicks were captured in their nests, handled for banding, measuring and weighing, and finally returned to their nest by the handler. Neighboring chicks from others species were not touched. During the course of each banding operation, both zones (disturbed and control) were visually monitored by an observer hidden inside a floating blind (Fig. 1) at a distance of 5 to 10 m and within a range of ten monitored nests. The behavior of the breeding adults of each zone was monitored. All banding operations were approved and supervised by the French Biological Research Center for Bird Populations (Centre de Recherches sur la Biologie des Populations d'Oiseaux, https://crbpo.mnhn.fr/).

## Breeding success monitoring

Within each zone (disturbed and control), all nests of Cattle Egrets, Little Egrets and Glossy Ibis were identified within 11 randomly selected Tamarisk trees. For each nest, we recorded the zone, the tree, the nesting species and the height of the nest in accordance with the following categories: <1 m above the water, 1–2 m above the water and >2 m above the water. Higher nests did not exceed 3m in height.

Nests were monitored from a mobile floating blind (Fig. 1) that allowed smooth movements of the observer over the entire colony without altering the behavior of the birds. With this method, it was possible to monitor the nests from a distance of approximately two meters. Monitoring took place two to three times a week, between 07h30 and 19h30, alternating morning and afternoon sessions. The first incursion to the site took place approximately two weeks after the colony was established and at least two weeks before the first banding operation took place (Fig. 2).

We considered that a nest survived between visits when at least one chick or egg was seen in the nest, or a breeder was attending the nest. A nest was considered to be abandoned when it was seen to be destroyed, empty or without an attending breeder over two successive sessions. The observer also noted the number of eggs and/or chicks seen in each nest and their respective age considering three age classes: 0–7 days old, 8–14 days old and 15–21 days old. As in other colonial waterbird species, classing of further ages was not possible because beyond the age of three weeks high mobility makes it difficult to attribute the chicks to a specific nest (*Frederick & Collopy, 1989a*; *Herring et al., 2010*).

## Statistical analysis

We assessed breeding success by means of the daily survival rate (DSR) of each nest (i.e., the probability that a nest was active at a certain point in time). In order to account for the higher probabilities of breeding success that are expected whenever a nest is monitored late in the season (due to the fact that the observer is not able to document failures occurred before the monitoring initiates; *Mayfield, 1975*; *Jones & Geupel, 2007*), we applied the logistic-exposure method developed for this purpose (*Shaffer, 2004*).

We considered the following fixed covariates as potential explanatory variables for the fate of a nest: *day* of the year (continuous), *height* of the nest in the tree (categorical with three levels: low, medium and high), *species* (categorical with three levels: Little Egret, Cattle Egret and Glossy Ibis), *zone* (two levels: control *vs.* disturbed), and *banding week* (the

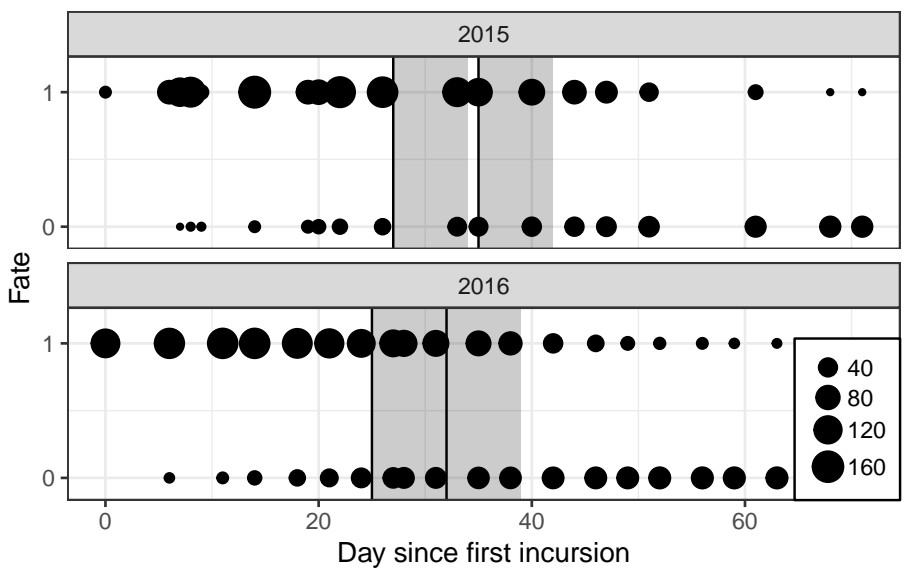

**Figure 2** **Number of nests monitored each year for each day of observation since the first incursion in the colony with a floating hide.** The fate ''1'' corresponds to a nest seen active and ''0'' to an abandoned nest. The two annual banding operations are presented by vertical black lines and the seven days following a banding operation by the dark grey font.

seven-day period following a banding operation, a categorical covariate with two levels). Additionally, we considered the following interactions of interest: *height* ×*species*, *height* ×*zone*, *species* ×*zone*, *zone* ×*banding week*, *banding week* ×*species*. In order to account for the dependency between the nests within a given tree, as well as the effect of year, we used the effect of *tree* (categorical with 11 levels each year) nested in that of *year* (categorical with two levels) as a random intercept. The linear predictor for our most general model was:

$$log_e\left(\frac{s_{ijk}}{1-s_{ijk}}\right) = Day_{ijk} + Height_{ijk} + Species_{ijk} + Zone_{ijk} + BandingWeek_{ijk} + Height_{ijk}$$
$$\times \, Species_{ijk} + Height_{ijk} \times Zone_{ijk} + Species_{ijk} \times Zone_{ijk} + Zone_{ijk}$$
$$\times \, BandingWeek_{ijk} + BandingWeek_{ijk} \times Species_{ijk} + Year_j + Tree_{k(j)}$$
$$Year_j \sim N\left(0, \sigma^2_{Year}\right)$$
$$Tree_{k(j)} \sim N\left(0, \sigma^2_{Tree}\right)$$

where $s_{ijk}$ is the daily survival rate of nest $i$ during year $j$ in tree $k$.

The package lme4 (*Bates et al., 2015*) in the software R (*R Core Team, 2017*) was used to fit the 144 candidate models that all had biological meaning. We performed automated model-selection routines using the package MuMIn (*Barton, 2018*) to rank the models by their AICc, the second-order variant of Akaike's Information Criteria (*Burnham & Anderson, 2002*). When a more in-depth examination of the effect of a variable was required, we tested for differences among the categories performing Tukey's post-hoc tests

using the multcomp package (*Hothorn, Bretz & Westfall, 2008*). Finally, the figures were drawn using the ggplot2 package (*Wickham, 2016*).

## RESULTS

Monitoring of breeders during the banding operations confirmed that in the control zone, breeders of all species (not only the Glossy Ibis) tended to remain in their nests, or at close proximity. Meanwhile, in the disturbed zone almost all breeders fled their nests and stayed away during the banding operation, only coming back after the banding operation was finished (all within 20 min after the team's retreat).

Over the two breeding seasons, 336 nests were monitored in the two zones, accounting for a total of 2,772 observations and a mean of 9.3 observations per nest (range: 2–19) until a success or a failure event was recorded.

Seven models for the estimation of daily survival rate were found within $\Delta$AICc <2 (Table 1). The seven models included an effect of the *day* in the season, with a DSR of nests showing a linear decrease throughout the breeding period ($\beta = -0.04 \pm 0.01$ SE, $z$ value $= -4.14$, *P*-value <0.001). All seven models also included an effect of *species* as well as the *height* of the nest above the water. DSR was found to be lowest for the nests situated less than 1m above the water as compared to those situated between 1 m and 2 m high (from the third model, the most parsimonious one with no interaction with species: $\beta = 1.08 \pm 0.31$ SE, $z$ value $= 3.46$, *P*-value $= 0.001$), or to those situated more than 2 m above the water ($\beta = 0.95 \pm 0.37$ SE, $z$ value $= 2.54$, *P*-value $= 0.01$). The interaction between *species* and *height* was present in four of the seven models. Results are illustrated in Fig. 3.

An adverse effect of banding operations for Glossy Ibis chicks was suggested by three of the seven preferred models, with lower DSR for Glossy ibises breeding in the disturbed zone as compared to the control zone, but a positive effect was given on the two others species (Fig. 4). Complementary models run from subsamples of the data from the disturbed zone and the control zone showed that lower survival rates for Glossy Ibis nests compared to Little Egret nests and Cattle Egret nests was found in the disturbed zone (Glossy Ibis nests vs. Cattle Egret nests: $\beta = -1.12 \pm 0.42$ SE, $z$ value $= -2.70$, *P*-value $= 0.02$ and Glossy Ibis nests vs. Little Egret nests: $\beta = -1.02 \pm 0.37$ SE, $z$ value $= -2.74$, *P*-value $= 0.02$) but not in the control zone (model with *species* variable was not retained as preferred model, see supplementary material). Nevertheless, three of the seven models refuted a temporal negative effect of the banding operations: DSR was found to be higher in the weeks following a banding operation (from fourth preferred model in Table 1: $\beta = 0.18 \pm 0.23$ SE, $z$ value $= 0.76.19$, *P*-value $= 0.45$).

## DISCUSSION

We found no evidence of major research-induced disturbances owing to regular banding operations on a mixed-species colony of breeding tree-nesting waterbirds. Factors most affecting the success of nests were the timing of breeding, with a decline in success over the season and the relative height at which nests were situated in the trees, with a reduced
**Table 1  Model selection table showing the ten best supported models to explain daily survival rate of Cattle Egrets, Little Egrets and Glossy Ibis nests in mixed-species heronry disturbed by banding operations in 2015 and 2016.** A total of 144 candidate models were tested, including the fixed effects of *species*, *day* since the first day of the survey, the position of the nest in a tree (*height*), *zone* in the colony (control *vs.* disturbed zone), the period of seven days following a disturbance (*banding week*), and second order interactions of interest. Year of survey and identification of the tree were considered as random variables for all candidate models. $K$ is the number of parameters in the model, $Log_e(L)$ is the value of the maximized log-likelihood function, $AIC_c$ is Akaike's Information Criterion adjusted for small samples, $\Delta AIC_c$ is the scaled value of $AIC_c$, and *weight* is the Akaike weight.

| Model | K | Loge(L) | AICc | delta | weight |
|---|---|---|---|---|---|
| Day+Height*Species+Species*Zone | 15 | −437.7 | 905.6 | 0.0 | 0.14 |
| Day+Height*Species | 12 | −441.1 | 906.2 | 0.6 | 0.10 |
| Day+Height+Species | 8 | −445.3 | 906.6 | 0.9 | 0.09 |
| Day+Height*Species+Species*Zone+BandingWeek | 16 | −437.4 | 907.1 | 1.4 | 0.07 |
| Day+Height+Species*Zone | 11 | −442.5 | 907.1 | 1.4 | 0.07 |
| Day+Height+Species+BandingWeek | 9 | −444.5 | 907.1 | 1.5 | 0.06 |
| Day+Height*Species+BandingWeek | 13 | −440.5 | 907.2 | 1.6 | 0.06 |
| Day+Height*Species+Zone | 13 | −440.9 | 908.0 | 2.3 | 0.04 |
| Day+Height+Species*Zone+BandingWeek | 12 | −442.0 | 908.1 | 2.5 | 0.04 |
| Day+Height*Species+Species*Zone+Zone*Banding Week | 17 | −437.1 | 908.4 | 2.8 | 0.03 |

survival for those that were lowest and closest to the water. Our results, however, did suggest a weak effect of research disturbance on the breeding success of Glossy Ibises, but no such effect was present in neighboring nests of Little or Cattle Egrets. Daily survival rates of Glossy Ibis nests was 4.0% lower in the zone that was disturbed by the operations as compared to the control zone; however, further examination revealed the effect lacked statistical significance.

Mechanisms underlying the occurrence of any potential negative effects provoked by research in this study, such as those observed in the Glossy Ibis, remain largely unclear. First, it is possible that any negative effects that are confined to this species are themselves a direct result of being the only species of the three that was handled and manipulated during the banding operations. However, we would have then expected to observe a reduction in the breeding success of this species over the days that followed each of the banding operations, something that did not occur (see the results from the models where the variable *banding week* was retained). This absence of effect was found by *Davis Jr & Parsons (1991)* in a comparison of the fate of repeated handling of Snowy Egret (*Egretta thula*) every two days prior to banding, with individuals only handled once when banded.

Alternatively, disturbance provoked by the operations may have indirectly affected the survival of nests of Glossy Ibis by either (i) increasing the nest abandonments, or (ii) increasing the risk of nest predation in the following days after the banding operation. Increased nest abandonments seem an unlikely cause given that on-site monitoring of breeders during the banding operations showed that all parents returned to their nests, independently of which species, within the 20 min following retreat of all participants from the colony.
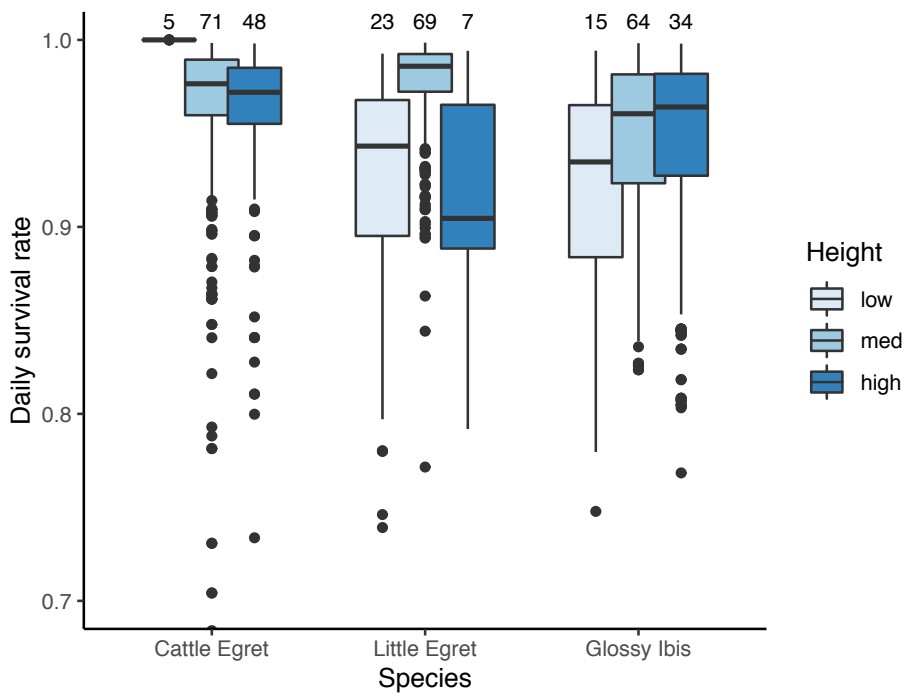

**Figure 3** **Effect of the height in tree on daily survival rate of nests of three species of tree-nesting waterbirds in a mixed-species heronry.** "low" denotes for nests located between the water level and 1 m, "med" for nests between 1 m and 2 m, and "high" for nests above 2 m. Estimates plotted in this figure are the predictions derived from the first model in Table 1. Sample sizes for each category are indicated at the top of the figure.

It is well known that nest predation can increase with human disturbance in other contexts, for instance along recreational trails (*Miller, Knight & Miller, 1998*). Similarly, in this study we cannot rule-out the possibility that the researcher's activities and disturbance inside the colony may have attracted aerial predators like the Marsh Harrier (*Circus aeruginosus*) or the Magpie (*Pica pica*). If this was the case however, we would thus have expected all three species to be similarly affected by the presence of such predators, and maybe even lower breeding success in higher nests; the opposite of what we found in this study.

A reduced survival rate for the nests that were low and close to the water was found. This was especially true for Glossy Ibis and Little Egret, but not for Cattle Egrets which only had 4% of their nests below 1 meter. It was unlikely that lower survival of lower nests was driven by the banding operations because there was no effect of the interaction *height ×zone*. Nevertheless, we cannot discard an effect of the floating blind, the observer being closer to lowest nests than the higher ones, potentially creating disturbance non detected by the observer. Further studies with a similar protocol but with increased levels of observations per week might allow confirming or discarding the effect of the floating blind.

Finally, we cannot discard that the power of the statistical tests was not sufficient to detect an adverse effect of the disturbance. The experimental protocol involved only one
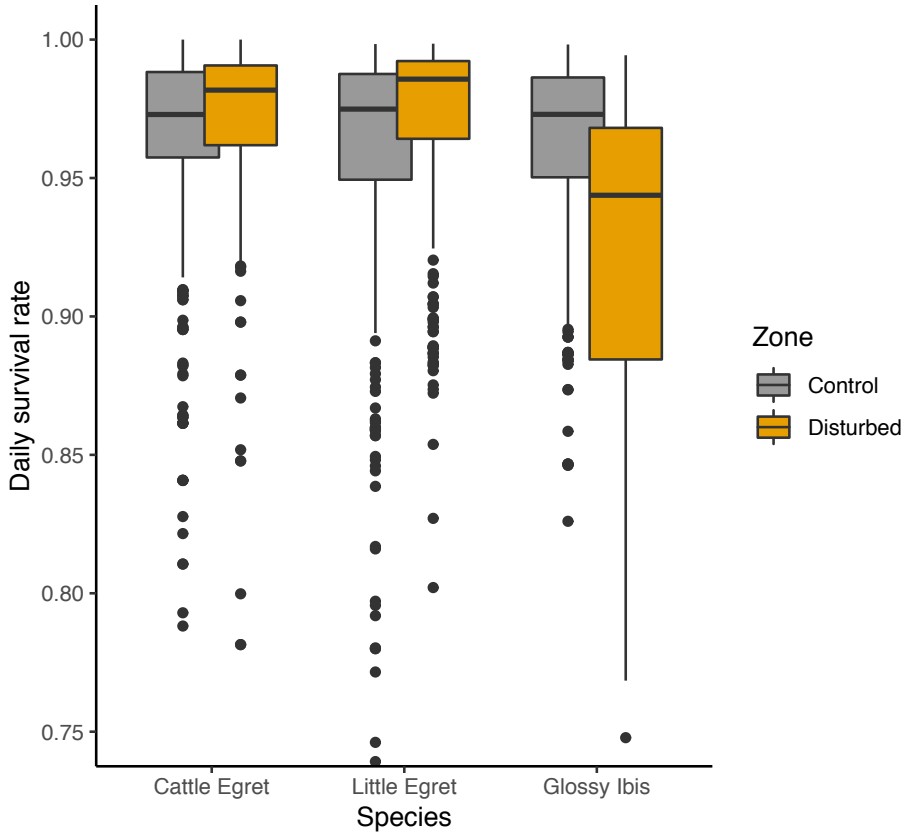

**Figure 4** **Effect of banding disturbance on daily survival rate of nests of three species of tree-nesting waterbirds in a mixed-species heronry.** Estimates plotted in this figure are the predictions derived from the first model in Table 1.

disturbed and one control zone each year, a quite limited amount of replicates. However, it is worth noting that we ruled out the possibility of a site effect because the position of the control site switched with the disturbed site between the two years.

Interestingly, nesting success slightly increased for Little Egret and Cattle Egret nests in the disturbed zone, where the nest success of Glossy Ibis tended to decrease. This result may be explained by the complex level of inter-specific interactions among the birds in the heronry's trees. Exclusion of Little Egret individuals by established Cattle Egrets has been shown by *Dami, Bennetts & Hafner (2006)*, although this occurred in the absence of Glossy Ibis individuals. Here it seems that the relatively lower breeding success of the Glossy Ibis, the primary species nesting in this colony, benefitted the Little Egret (+1.7% in DSR), and to a lesser extent, the Cattle Egret (+0.8% in DSR).

In this study, we carried out an experimental disturbance treatment of low magnitude and frequency (i.e., two banding operations limited to one hour per breeding season). This protocol is in accordance with the one developed in 1982 by expert ornithologists (*Hoffmann, Hafner & Salathe, 1996*). Nevertheless, it can be considered as low disturbance compared to other studies such as that of *Tremblay & Ellison (1979)* on Black-crowned

Night Heron which included up to 15 visits of 45 min to 4 h each. Such major disturbances were found to discourage new breeders, increase nest abandonment in particular during egg laying periods and increase nestling mortality.

The experimental protocol and the statistical treatment that we adopted for this study enabled us to test the effects of specific temporal events such as occurrence of research-related disturbance on the estimation of breeding success simultaneously on three different species of tree-nesting waterbirds. Contrary to more typical estimates like the number of fledglings per nest, DSR allowed us to test for the effect of a temporal event. Specifically, in our study, adding a temporal covariate *BandingWeek* allowed us to discard the effect of handling on the survival of Glossy Ibis chicks.

## CONCLUSIONS

Over the last century, research disturbance in ornithology has mainly focused on waterbirds (*Goering & Cherry, 1971*; *Frederick & Collopy, 1989b*; *Davis Jr & Parsons, 1991*; *Götmark, 1992*; *Bowman et al., 1994*; *Kuiken et al., 1997*; *Carney & Sydeman, 1999*; *Nisbet, 2000*). Nevertheless, Passeriformes were over-represented in the meta-analysis on research disturbance by *Ibáñez Álamo, Sanllorente & Soler (2012)* because they considered that most of the studies involving waterbird species (and all comprising Pelicaniformes) "included only anecdotal evidence or were not properly designed to investigate this topic". It is clear that in order to overcome this limitation, and also because contrasting results may arise depending on the taxonomic group examined, highly effective experimental studies for waterbirds are needed. This study contributes to this with the numerical assessment of research disturbance on waterbirds.

In this study we showed at least partial evidence for negligible effects of banding operations on colonial tree-nesting waterbirds. Banding of birds for the purposes of population monitoring and individual identification is an ubiquitous ornithological practice that has proven extremely helpful to assess demographic parameters of countless wild bird populations (*Santoro, Green & Figuerola, 2016*; *Champagnon et al., 2018*). However, before committing resources and substantial efforts to start using them, it is important to not only assess issues like whether the fitted devices lead to reliable demographic parameters, but also that the banding operations themselves do not compromise survival and reproduction in any meaningful way (*Saraux et al., 2011*; *Spotswood et al., 2012*; *Griesser et al., 2012*; *Guillemain et al., 2015a*; *Guillemain et al., 2015b*; *Weiser et al., 2016*; *Stein et al., 2017*; *Avila-Villegas, 2018*; *Border et al., 2018*). We advocate that any study involving manipulation of animals should consider potential adverse effects based on the literature first. Animal care and ethics committees, as well as national banding organizations that delivered permits for scientific studies involving wild animals, participate in updated guidelines that take into account the kind of study provided here (*Casper, 2009*; *Fair, Paul & Jones, 2010*; *Vitale et al., 2018*).

## ACKNOWLEDGEMENTS

We warmly thank Lisa Ernoul and Oscar Sanchez Macouzet for improving the language and useful comments made on a earlier version of the manuscript. We are indebted to Thomas Blanchon, Yves Kayser, Jérémiah Petit, and Rémi Tiné for their advice during the design and the implementation of the experiment. We thank the Syndicat Mixte pour la protection et la gestion de la Camargue Gardoise for giving access to the nature reserve. We thank all the participants that participated to banding operations of Glossy Ibis since 2006.

### Funding

This work was supported by Fondation Tour du Valat. The funders had no role in study design, data collection and analysis, decision to publish, or preparation of the manuscript.

### Grant Disclosures

The following grant information was disclosed by the authors:
Fondation Tour du Valat.

### Competing Interests

The authors declare there are no competing interests.

### Author Contributions

- Jocelyn Champagnon conceived and designed the experiments, analyzed the data, contributed reagents/materials/analysis tools, prepared figures and/or tables, authored or reviewed drafts of the paper, approved the final draft.
- Hugo Carré and Lisa Gili performed the experiments, analyzed the data, authored or reviewed drafts of the paper, approved the final draft.

### Animal Ethics

The following information was supplied relating to ethical approvals (i.e., approving body and any reference numbers):

All banding operations were conducted under the program ''Dynamique de la population d'Ibis falcinelle *Plegadis falcinellus* en Camargue'' approved by the Centre de Recherches sur la Biologie des Populations d'Oiseaux.

### Field Study Permissions

The following information was supplied relating to field study approvals (i.e., approving body and any reference numbers):

Field experiments were conducted within the réserve naturelle régionale du Scamandre and approved by le Syndicat Mixte pour la protection et la gestion de la Camargue Gardoise.

### Data Availability

Script and data are available at figshare: Champagnon J.; Gili L.; Carré H. (2019): Effects of Research Disturbance on Nest Survival in a Mixed Colony of Waterbirds. Figshare online resource. https://doi.org/10.6084/m9.figshare.8942267.v1.

## Supplemental Information

Supplemental information for this article can be found online at http://dx.doi.org/10.7717/peerj.7844#supplemental-information.

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
