# Peer review of "Effects of research disturbance on nest survival in a mixed colony of waterbirds"

_PeerJ, doi:10.7717/peerj.7844_

## Round 0.1 · original submission · Major Revisions

Two reviewers have thoroughly reviewed your manuscript and have found your study to be a valuable contribution. Both reviewers indicated a number of points that need clarification, additional discussion, or minor additional analysis - you should address all of these points carefully.

Please note that both reviewers took the time to look through your data and statistical code and paid careful attention to your statistical methods.

The reviewers also highlighted instances where the writing is unclear or hard to understand. Please ensure the next draft of the manuscript is more concise and clearer.

I agree with reviewer 1 that more tables / figures would improve this manuscript and it's clarity. In addition to data figures, perhaps the authors could include a photograph of their blind, heronry, and a band?

Reviewer 2 makes an important point that disturbance in this study is relatively gentle compared to other studies. It is important for you to address this in your discussion, particularly for motivating future research.

I have several points of my own that I would like to see addressed.

Lines 160-162 & 168-173: The p-values are above a typical 0.05. Do you considered these statistically and biologically significant differences? Given figure 2, the effects of disturbance on egrets seems minimal. The second preferred model indicates a species*zone interaction and the plot is pretty clear that this interaction suggests that ibis in the disturbed zone have reduced survival but survival looks pretty similar for all three species in the undisturbed zone. This would mean that, in your results section, where you imply ibis had lower survival than the other species is not generally correct. Ibis have lower survival than the other two species in the disturbed zone but not in the control zone. Given the significance of this interaction, perhaps the authors would consider running a single separate model each for the disturbed and undisturbed zones only to evaluate species-level differences. It is common to further explore models this way to accurately interpret interactions. In line 185 you say further analysis found the effect of disturbance on ibis to be non-significant but it's unclear to me where this analysis is explicitly.

I think it would be valuable to define "ringing" in your abstract and early on in your introduction. For those who are not ornithologists, "ringing" may be easily misconstrued as ringing the necks and euthanizing these birds. Clearly this is not what you mean. To reach as broad of a conservation audience as possible, it would be useful to define this term for non ornithologists.

Additionally, it might be nice to briefly discuss the implications of your findings for agencies or animal care and use committees at institutions. Beyond your study advocating that other researchers consider and look into the effects of their studies on their organisms, should agencies or animal care and use committees more explicitly consider the evidence for the effects of research on study animals?

It's also useful to note that this is an n=1 study. Your study only had one control and one impact replicate each. This does not reduce the value of your study but perhaps there are simply differences between the two study sites that are not attributable to researcher disturbance. Please add a discussion of this in your next draft.

Reviewer 1 ·

Basic reporting

The manuscript is logically and clearly written and provides sufficient background information to be informative. The study covers a clear and important question.

Experimental design

The experimental design is clearly detailed.

There are two areas where I would appreciate further detail. It was unclear to me what was meant by the 'ringing' variable on line 133. Please add a more detailed description.

Based on the text it was unclear to me how the 72 candidate models were developed, by digging into the code I see that the dredge() function was used, a function that does automated model selection. This should be detailed in the methods, along with an explanation of why this method was chosen, instead of creating a smaller set of candidate models a priori.

Validity of the findings

The findings are solid, and are an important investigation of a large assumption that is made frequently in field ornithology. I think the communication of those results could be improved with additional figures/tables, but I acknowledge that is personal preference to some degree. I personally find paragraphs of text with parenthetical reporting of parameters to be more difficult to synthesize.

Additional comments

The term waterbirds is used in this manuscript, which is a term used by different folks to mean different groups of species. Please define what you mean by waterbirds at its first use.

I reviewed the code, a few comments, overall its good.

Remove the setwd() commands from the top of the code, those are specific to your machine, and will not work on anyone else's

Please move all the library() commands to the top of the script, so that the user can ensure they have all the necessary packages installed/loaded at the beginning

There are several places throughout the script where comments are made (which is good), but they are not in English. My understanding is that PeerJ requires them all to be in English. Please carefully review the code and update those comments.

Reviewer 2 ·

Basic reporting

The English language should be improved to increase the article's clarity. The current phrasing of certain sentences make comprehension difficult. For example, the sentences starting on lines 39, 57, 60, and 62 could be improved. I suggest following IOU guidelines whenever using common names, capitalize ‘Ibis’ and ‘Egret’ https://www.worldbirdnames.org/bow/pelicans/

A detailed field background supported with strong references is provided.

The article adheres to the professional standard of current scientific articles. The raw data is present and well organized, but species names and heights are not able to be interpreted in the raw data (codes are used without a key). Figure 1 was difficult to interpret. The clarity of this figure could be improved by including a description of what the black circles represent. I was also confused as to how the number of nests that were abandoned appears to decrease over the nesting season. Can an abandoned nest change it's fate? If so, how? Some of the annotations in the R code are in French only.

The hypothesis is clearly outlined, tested, and discussed using the data collected.

Experimental design

The research question is well defined, relevant and meaningful. A knowledge gap, does disturbance resulting from research impact the daily survival of colony nesting birds, is identified and addressed in this article.

The investigation is rigorous and followed a large number of nests throughout two nesting seasons, following a high technical and ethical standard.

The methods are lacking a detailed description of the floating blinds used to monitor nests. The article did not make it clear if the floating blinds were mobile or stationary. The citation provided (Shugart et al 1981) in regards to the floating blind pertains to tunnels used to enter blinds. The construction, size, and mobility of the blinds used in the article are unknown. Shugart et al 1981 notes that the methods used resulted in altering the focal species behavior. The article in review does not address this.

The statistical methods used are not fully described in the article. A Tukey's post hoc test was used in the R code, but not outlined in the statistical analysis section. The R packages used in the statistical analysis are not presented or cited in the article. Program R can be cited using the code "citation()", while packages within R can be cited using the code "citation(package = "PACKAGENAME")" for example "citation(package = "lme4")".

Validity of the findings

Resulted are accurately presented.

All underlying data have been provided, they are robust, statistically sound, and controlled. I suggest an adjustment to the statistical analysis that is outlined in general comments.

Conclusions are linked to original research question. The authors thoroughly describe the ringing methods that cause the disturbance in the methods section, but I would suggest that this is reiterated in the conclusion. The ringing methods used focused on minimizing disturbance, which resulted in a total of 2 research related disturbance events totaling 2 hours (or less) over the entire nesting period. The authors second model (presented in figure 2) found that, while not statistically significant, there was a signal that DSR decreased in the disturbed area. Other studies may not implement such a well managed, and limited amount of disturbance, which could result in a larger impact to DSR. While this is speculation, it highlights the fact that disturbance relatively minor in this study and it was tested as a binomial factor.

Additional comments

The overall structure of the paper is great. The paper flows logically, ideas are presented and supported with literature. The article does a good job of outlining knowledge gaps and presents how it aims to address them. Improving the language will substantially increase a reader's ability to interpret the article. The statistical analysis is good, but it could benefit from some additional explanation. The article would benefit greatly from an improved description of the floating blind.

It is interesting that the top five models pulled nest height as an important factor in DSR. A quick investigation into nest height by species showed that species 'AG' is responsible for a majority of the nests at height 'a', but this species rarely nests at height 'c'. Similarly, species 'GB' is responsible for the majority of nests at height 'c'. I would suggest adding 'height * species' as an additional interaction effect into the model. Including this interaction would allow for the effect of height and species on DSR to be better understood.

Is it possible that ringing events and/or nest observations from the floating blind disproportionately effected nests nearest the water?

---

## Round 0.2 · accepted · Accept

Thank you for taking the time to carefully respond to each of the reviewer's comments and my comments as well. The manuscript clarity is much improved and the analysis and figure updates improve the quality of the study and its presentation.

Congratulations on the acceptance of a very nice study.